# Universal Correspondence Network

**Christopher B. Choy**
Stanford University
chrischoy@ai.stanford.edu

**JunYoung Gwak**
Stanford University
jgwak@ai.stanford.edu

**Silvio Savarese**
Stanford University
ssilvio@stanford.edu

**Manmohan Chandraker**
NEC Laboratories America, Inc.
manu@nec-labs.com

## Abstract

We present a deep learning framework for accurate visual correspondences and demonstrate its effectiveness for both geometric and semantic matching, spanning across rigid motions to intra-class shape or appearance variations. In contrast to previous CNN-based approaches that optimize a surrogate patch similarity objective, we use deep metric learning to directly learn a feature space that preserves either geometric or semantic similarity. Our fully convolutional architecture, along with a novel correspondence contrastive loss allows faster training by effective reuse of computations, accurate gradient computation through the use of thousands of examples per image pair and faster testing with $O(n)$ feed forward passes for $n$ keypoints, instead of $O(n^2)$ for typical patch similarity methods. We propose a convolutional spatial transformer to mimic patch normalization in traditional features like SIFT, which is shown to dramatically boost accuracy for semantic correspondences across intra-class shape variations. Extensive experiments on KITTI, PASCAL, and CUB-2011 datasets demonstrate the significant advantages of our features over prior works that use either hand-constructed or learned features.

## 1   Introduction

Correspondence estimation is the workhorse that drives several fundamental problems in computer vision, such as 3D reconstruction, image retrieval or object recognition. Applications such as structure from motion or panorama stitching that demand sub-pixel accuracy rely on sparse keypoint matches using descriptors like SIFT [22]. In other cases, dense correspondences in the form of stereo disparities, optical flow or dense trajectories are used for applications such as surface reconstruction, tracking, video analysis or stabilization. In yet other scenarios, correspondences are sought not between projections of the same 3D point in different images, but between semantic analogs across different instances within a category, such as beaks of different birds or headlights of cars. Thus, in its most general form, the notion of visual correspondence estimation spans the range from low-level feature matching to high-level object or scene understanding.

Traditionally, correspondence estimation relies on hand-designed features or domain-specific priors. In recent years, there has been an increasing interest in leveraging the power of convolutional neural networks (CNNs) to estimate visual correspondences. For example, a Siamese network may take a pair of image patches and generate their similarity as the output [1, 34, 35]. Intermediate convolution layer activations from the above CNNs are also usable as generic features.

However, such intermediate activations are not optimized for the visual correspondence task. Such features are trained for a surrogate objective function (patch similarity) and do not necessarily form a metric space for visual correspondence and thus, any metric operation such as distance does not have

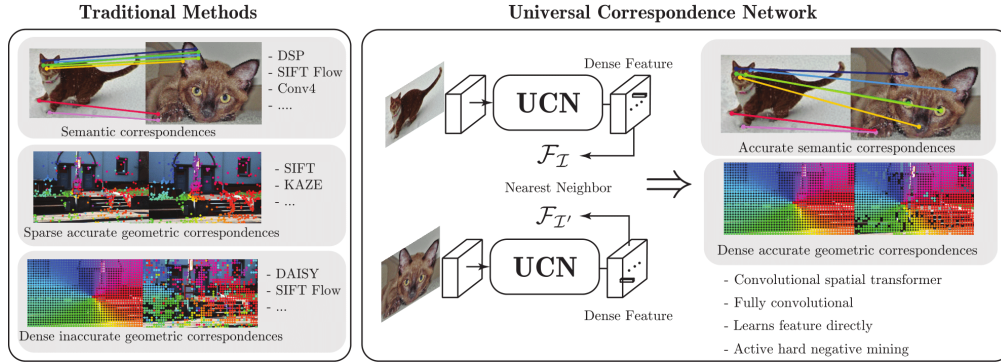

Figure 1: Various types of correspondence problems have traditionally required different specialized methods: for example, SIFT or SURF for sparse structure from motion, DAISY or DSP for dense matching, SIFT Flow or FlowWeb for semantic matching. The Universal Correspondence Network accurately and efficiently learns a metric space for geometric correspondences, dense trajectories or semantic correspondences.

explicit interpretation. In addition, patch similarity is inherently inefficient, since features have to be extracted even for overlapping regions within patches. Further, it requires $O(n^2)$ feed-forward passes to compare each of $n$ patches with $n$ other patches in a different image.

In contrast, we present the Universal Correspondence Network (UCN), a CNN-based generic discriminative framework that learns both geometric and semantic visual correspondences. Unlike many previous CNNs for patch similarity, we use deep metric learning to directly learn the mapping, or feature, that preserves similarity (either geometric or semantic) for generic correspondences. The mapping is, thus, invariant to projective transformations, intra-class shape or appearance variations, or any other variations that are irrelevant to the considered similarity. We propose a novel *correspondence contrastive loss* that allows faster training by efficiently sharing computations and effectively encoding neighborhood relations in feature space. At test time, correspondence reduces to a nearest neighbor search in feature space, which is more efficient than evaluating pairwise patch similarities.

The UCN is fully convolutional, allowing efficient generation of dense features. We propose an on-the-fly *active hard-negative mining* strategy for faster training. In addition, we propose a novel adaptation of the spatial transformer [13], called the *convolutional spatial transformer*, desgined to make our features invariant to particular families of transformations. By learning the optimal feature space that compensates for affine transformations, the convolutional spatial transformer imparts the ability to mimic patch normalization of descriptors such as SIFT. Figure 1 illustrates our framework.

The capabilities of UCN are compared to a few important prior approaches in Table 1. Empirically, the correspondences obtained from the UCN are denser and more accurate than most prior approaches specialized for a particular task. We demonstrate this experimentally by showing state-of-the-art performances on sparse SFM on KITTI, as well as dense geometric or semantic correspondences on both rigid and non-rigid bodies in KITTI, PASCAL and CUB datasets.

To summarize, we propose a novel end-to-end system that optimizes a general correspondence objective, independent of domain, with the following main contributions:

- Deep metric learning with an efficient correspondence constrastive loss for learning a feature representation that is *optimized for the given correspondence task*.
- Fully convolutional network for dense and efficient feature extraction, along with *fast active hard negative mining*.
- Fully convolutional spatial transformer for *patch normalization*.
- State-of-the-art correspondences across sparse SFM, dense matching and semantic matching, encompassing rigid bodies, non-rigid bodies and intra-class shape or appearance variations.

## 2   Related Works

**Correspondences**   Visual features form basic building blocks for many computer vision applications. Carefully designed features and kernel methods have influenced many fields such as structure

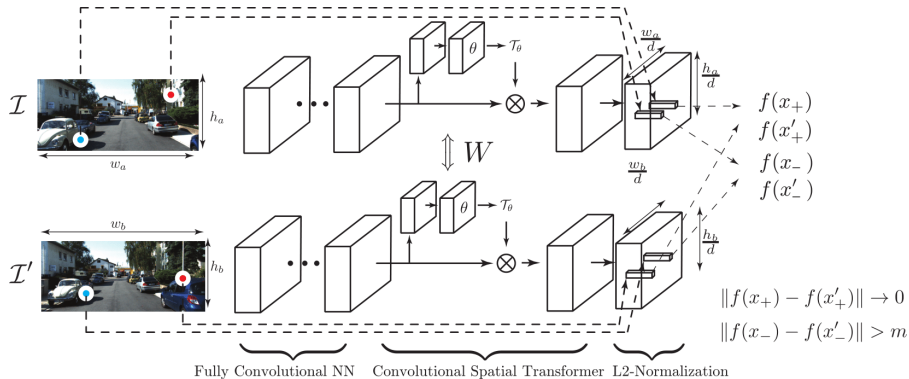

$$\|f(x_+) - f(x'_+)\| \to 0$$
$$\|f(x_-) - f(x'_-)\| > m$$

Fully Convolutional NN    Convolutional Spatial Transformer    L2-Normalization

Figure 2: System overview: The network is fully convolutional, consisting of a series of convolutions, pooling, nonlinearities and a convolutional spatial transformer, followed by channel-wise L2 normalization and correspondence contrastive loss. As inputs, the network takes a pair of images and coordinates of corresponding points in these images (blue: positive, red: negative). Features that correspond to the positive points (from both images) are trained to be closer to each other, while features that correspond to negative points are trained to be a certain margin apart. Before the last L2 normalization and after the FCNN, we placed a convolutional spatial transformer to normalize patches or take larger context into account.

| Features | Dense | Geometric Corr. | Semantic Corr. | Trainable | Efficient | Metric Space |
|---|---|---|---|---|---|---|
| SIFT [22] | ✗ | ✓ | ✗ | ✗ | ✓ | ✗ |
| DAISY [28] | ✓ | ✓ | ✗ | ✗ | ✓ | ✗ |
| Conv4 [21] | ✓ | ✗ | ✓ | ✓ | ✓ | ✗ |
| DeepMatching [25] | ✓ | ✓ | ✗ | ✗ | ✗ | ✓ |
| Patch-CNN [34] | ✓ | ✓ | ✗ | ✓ | ✗ | ✗ |
| LIFT [33] | ✗ | ✓ | ✗ | ✓ | ✓ | ✓ |
| Ours | ✓ | ✓ | ✓ | ✓ | ✓ | ✓ |

Table 1: Comparison of prior state-of-the-art methods with UCN (ours). The UCN generates dense and accurate correspondences for either geometric or semantic correspondence tasks. The UCN directly learns the feature space to achieve high accuracy and has distinct efficiency advantages, as discussed in Section 3.

from motion, object recognition and image classification. Several hand-designed features, such as SIFT, HOG, SURF and DAISY have found widespread applications [22, 3, 28, 8].

Recently, many CNN-based similarity measures have been proposed. A Siamese network is used in [34] to measure patch similarity. A driving dataset is used to train a CNN for patch similarity in [1], while [35] also uses a Siamese network for measuring patch similarity for stereo matching. A CNN pretrained on ImageNet is analyzed for visual and semantic correspondence in [21]. Correspondences are learned in [16] across both appearance and a global shape deformation by exploiting relationships in fine-grained datasets. In contrast, we learn a metric space in which metric operations have direct interpretations, rather than optimizing the network for patch similarity and using the intermediate features. For this, we implement a fully convolutional architecture with a correspondence contrastive loss that allows faster training and testing and propose a convolutional spatial transformer for local patch normalization.

**Metric learning using neural networks**    Neural networks are used in [5] for learning a mapping where the Euclidean distance in the space preserves semantic distance. The loss function for learning similarity metric using Siamese networks is subsequently formalized by [7, 12]. Recently, a triplet loss is used by [29] for fine-grained image ranking, while the triplet loss is also used for face recognition and clustering in [26]. Mini-batches are used for efficiently training the network in [27].

**CNN invariances and spatial transformations**    A CNN is invariant to some types of transformations such as translation and scale due to convolution and pooling layers. However, explicitly handling such invariances in forms of data augmentation or explicit network structure yields higher accuracy in many tasks [17, 15, 13]. Recently, a spatial transformer network is proposed in [13] to learn how to zoom in, rotate, or apply arbitrary transformations to an object of interest.

**Fully convolutional neural network**    Fully connected layers are converted in $1 \times 1$ convolutional filters in [20] to propose a fully convolutional framework for segmentation. Changing a regular CNN to a fully convolutional network for detection leads to speed and accuracy gains in [11]. Similar to these works, we gain the efficiency of a fully convolutional architecture through reusing activations

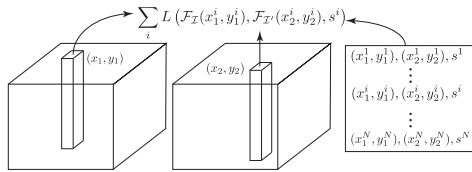

Figure 3: Correspondence contrastive loss takes three inputs: two dense features extracted from images and a correspondence table for positive and negative pairs.

| Methods | # examples per image pair | # feed forwards per test |
|---|---|---|
| Siamese Network | 1 | $O(N^2)$ |
| Triplet Loss | 2 | $O(N)$ |
| Contrastive Loss | 1 | $O(N)$ |
| Corres. Contrast. Loss | $> 10^3$ | $O(N)$ |

Table 2: Comparisons between metric learning methods for visual correspondence. Feature learning allows faster test times. Correspondence contrastive loss allows us to use many more correspondences in one pair of images than other methods.

for overlapping regions. Further, since number of training instances is much larger than number of images in a batch, variance in the gradient is reduced, leading to faster training and convergence.

## 3   Universal Correspondence Network

We now present the details of our framework. Recall that the Universal Correspondence Network is trained to directly learn a mapping that preserves similarity instead of relying on surrogate features. We discuss the fully convolutional nature of the architecture, a novel correspondence contrastive loss for faster training and testing, active hard negative mining, as well as the convolutional spatial transformer that enables patch normalization.

**Fully Convolutional Feature Learning**   To speed up training and use resources efficiently, we implement fully convolutional feature learning, which has several benefits. First, the network can reuse some of the activations computed for overlapping regions. Second, we can train several thousand correspondences for each image pair, which provides the network an accurate gradient for faster learning. Third, hard negative mining is efficient and straightforward, as discussed subsequently. Fourth, unlike patch-based methods, it can be used to extract dense features efficiently from images of arbitrary sizes.

During testing, the fully convolutional network is faster as well. Patch similarity based networks such as [1, 34, 35] require $O(n^2)$ feed forward passes, where $n$ is the number of keypoints in each image, as compared to only $O(n)$ for our network. We note that extracting intermediate layer activations as a surrogate mapping is a comparatively suboptimal choice since those activations are not directly trained on the visual correspondence task.

**Correspondence Contrastive Loss**   Learning a metric space for visual correspondence requires encoding corresponding points (in different views) to be mapped to neighboring points in the feature space. To encode the constraints, we propose a generalization of the contrastive loss [7, 12], called *correspondence contrastive loss*. Let $\mathcal{F}_{\mathcal{I}}(\mathbf{x})$ denote the feature in image $\mathcal{I}$ at location $\mathbf{x} = (x, y)$. The loss function takes features from images $\mathcal{I}$ and $\mathcal{I}'$, at coordinates $\mathbf{x}$ and $\mathbf{x}'$, respectively (see Figure 3). If the coordinates $\mathbf{x}$ and $\mathbf{x}'$ correspond to the same 3D point, we use the pair as a positive pair that are encouraged to be close in the feature space, otherwise as a negative pair that are encouraged to be at least margin $m$ apart. We denote $s = 1$ for a positive pair and $s = 0$ for a negative pair. The full correspondence contrastive loss is given by

$$L = \frac{1}{2N} \sum_i^N s_i \|\mathcal{F}_{\mathcal{I}}(\mathbf{x}_i) - \mathcal{F}_{\mathcal{I}'}(\mathbf{x}_i')\|^2 + (1 - s_i) \max(0, m - \|\mathcal{F}_{\mathcal{I}}(\mathbf{x}) - \mathcal{F}_{\mathcal{I}'}(\mathbf{x}_i')\|)^2 \quad (1)$$

For each image pair, we sample correspondences from the training set. For instance, for KITTI dataset, if we use each laser scan point, we can train up to 100k points in a single image pair. However in practice, we used 3k correspondences to limit memory consumption. This allows more accurate gradient computations than traditional contrastive loss, which yields one example per image pair. We again note that the number of feed forward passes at test time is $O(n)$ compared to $O(n^2)$ for Siamese network variants [1, 35, 34]. Table 2 summarizes the advantages of a fully convolutional architecture with correspondence contrastive loss.

**Hard Negative Mining**   The correspondence contrastive loss in Eq. (1) consists of two terms. The first term minimizes the distance between positive pairs and the second term pushes negative pairs to be at least margin $m$ away from each other. Thus, the second term is only active when the distance between the features $\mathcal{F}_{\mathcal{I}}(\mathbf{x}_i)$ and $\mathcal{F}_{\mathcal{I}'}(\mathbf{x}_i')$ are smaller than the margin $m$. Such boundary defines the

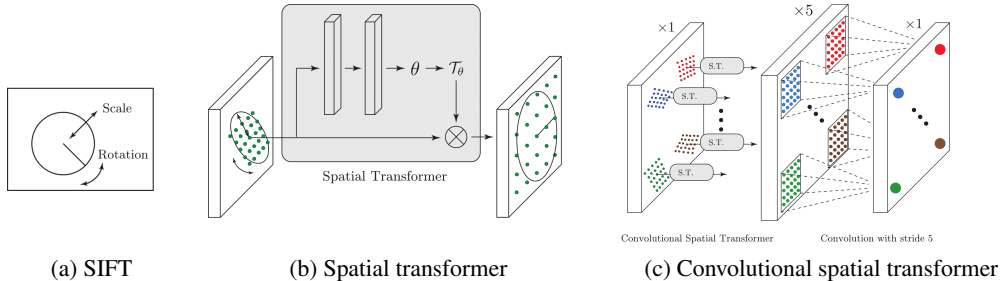

|            |                        |                                     |
|------------|------------------------|-------------------------------------|
| (a) SIFT   | (b) Spatial transformer | (c) Convolutional spatial transformer |

Figure 4: (a) SIFT normalizes for rotation and scaling. (b) The spatial transformer takes the whole image as an input to estimate a transformation. (c) Our convolutional spatial transformer applies an independent transformation to features.

metric space, so it is crucial to find the negatives that violate the constraint and train the network to push the negatives away. However, random negative pairs do not contribute to training since they are are generally far from each other in the embedding space.

Instead, we actively mine negative pairs that violate the constraints the most to dramatically speed up training. We extract features from the first image and find the nearest neighbor in the second image. If the location is far from the ground truth correspondence location, we use the pair as a negative. We compute the nearest neighbor for all ground truth points on the first image. Such mining process is time consuming since it requires $O(mn)$ comparisons for $m$ and $n$ feature points in the two images, respectively. Our experiments use a few thousand points for $n$, with $m$ being all the features on the second image, which is as large as 22000. We use a GPU implementation to speed up the K-NN search [10] and embed it as a Caffe layer to actively mine hard negatives on-the-fly.

**Convolutional Spatial Transformer**   CNNs are known to handle some degree of scale and rotation invariances. However, handling spatial transformations explicitly using data-augmentation or a special network structure have been shown to be more successful in many tasks [13, 15, 16, 17]. For visual correspondence, finding the right scale and rotation is crucial, which is traditionally achieved through *patch normalization* [23, 22]. A series of simple convolutions and poolings cannot mimic such complex spatial transformations.

To mimic patch normalization, we borrow the idea of the spatial transformer layer [13]. However, instead of a global image transformation, each keypoint in the image can undergo an independent transformation. Thus, we propose a convolutional version to generate the transformed activations, called the *convolutional spatial transformer*. As demonstrated in our experiments, this is especially important for correspondences across large intra-class shape variations.

The proposed transformer takes its input from a lower layer and for each output feature, applies an independent spatial transformation. The transformation parameters are also extracted convolutionally. Since they go through an independent transformation, the transformed activations are placed inside a larger activation without overlap and then go through a successive convolution with the stride to combine the transformed activations independently. The stride size has to be equal to the size of the spatial transformer kernel size. Figure 4 illustrates the convolutional spatial transformer module.

## 4   Experiments

We use Caffe [14] package for implementation. Since it does not support the new layers we propose, we implement the correspondence contrastive loss layer and the convolutional spatial transformer layer, the K-NN layer based on [10] and the channel-wise L2 normalization layer. We did not use flattening layer nor the fully connected layer to make the network fully convolutional, generating features at every fourth pixel. For accurate localization, we then extract features densely using bilinear interpolation to mitigate quantization error for sparse correspondences. Please refer to the supplementary materials for the network implementation details and visualization.

For each experiment setup, we train and test three variations of networks. First, the network has hard negative mining and spatial transformer (Ours-HN-ST). Second, the same network without spatial transformer (Ours-HN). Third, the same network without spatial transformer and hard negative mining, providing random negative samples that are at least certain pixels apart from the ground

| method | SIFT-NN [22] | HOG-NN [8] | SIFT-flow [19] | DaisyFF [31] | DSP [18] | DM best (½) [25] | Ours-HN | Ours-HN-ST |
|---|---|---|---|---|---|---|---|---|
| MPI-Sintel | 68.4 | 71.2 | 89.0 | 87.3 | 85.3 | 89.2 | **91.5** | 90.7 |
| KITTI | 48.9 | 53.7 | 67.3 | 79.6 | 58.0 | 85.6 | **86.5** | 83.4 |

Table 3: Matching performance PCK@10px on KITTI Flow 2015 [24] and MPI-Sintel [6]. Note that DaisyFF, DSP, DM use global optimization whereas we only use the raw correspondences from nearest neighbor matches.

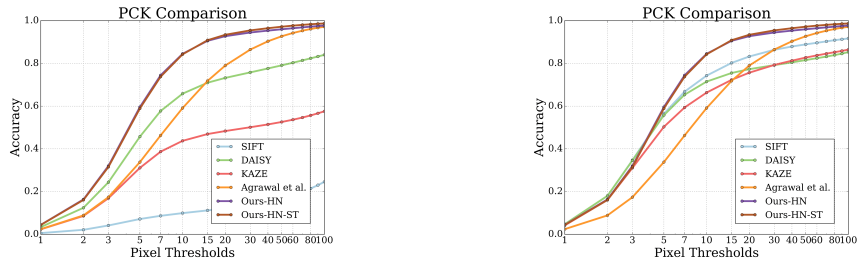

(a) PCK performance for dense features NN                (b) PCK performance on keypoints NN

Figure 5: Comparison of PCK performance on KITTI raw dataset (a) PCK performance of the densely extracted feature nearest neighbor (b) PCK performance for keypoint features nearest neighbor and the dense CNN feature nearest neighbor

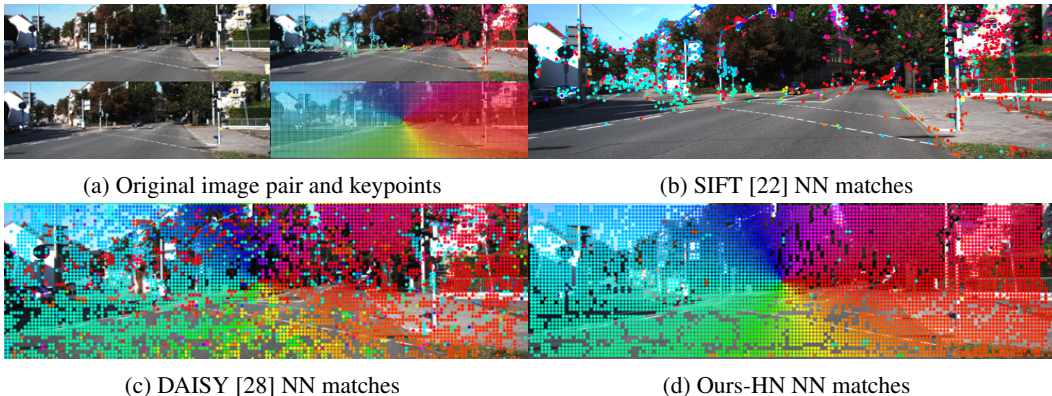

(a) Original image pair and keypoints                (b) SIFT [22] NN matches

(c) DAISY [28] NN matches                (d) Ours-HN NN matches

Figure 6: Visualization of nearest neighbor (NN) matches on KITTI images (a) from top to bottom, first and second images and FAST keypoints and dense keypoints on the first image (b) NN of SIFT matches on second image. (c) NN of dense DAISY matches on second image. (d) NN of our dense UCN matches on second image.

truth correspondence location instead (Ours-RN). With this configuration of networks, we verify the effectiveness of each component of Universal Correspondence Network.

**Datasets and Metrics**    We evaluate our UCN on three different tasks: geometric correspondence, semantic correspondence and accuracy of correspondences for camera localization. For geometric correspondence (matching images of same 3D point in different views), we use two optical flow datasets from KITTI 2015 Flow benchmark and MPI Sintel dataset and split their training set into a training and a validation set individually. The exact splits are available on the project website. alidation For semantic correspondences (finding the same functional part from different instances), we use the PASCAL-Berkeley dataset with keypoint annotations [9, 4] and a subset used by FlowWeb [36]. We also compare against prior state-of-the-art on the Caltech-UCSD Bird dataset[30]. To test the accuracy of correspondences for camera motion estimation, we use the raw KITTI driving sequences which include Velodyne scans, GPS and IMU measurements. Velodyne points are projected in successive frames to establish correspondences and any points on moving objects are removed.

To measure performance, we use the percentage of correct keypoints (PCK) metric [21, 36, 16] (or equivalently "accuracy@T" [25]). We extract features densely or on a set of sparse keypoints (for semantic correspondence) from a query image and find the nearest neighboring feature in the second image as the predicted correspondence. The correspondence is classified as correct if the predicted keypoint is closer than $T$ pixels to ground-truth (in short, PCK@$T$). Unlike many prior works, we do not apply any post-processing, such as global optimization with an MRF. This is to capture the performance of raw correspondences from UCN, which already surpasses previous methods.

**Geometric Correspondence**    We pick random 1000 correspondences in each KITTI or MPI Sintel image during training. We consider a correspondence as a hard negative if the nearest neighbor in

| | aero | bike | bird | boat | bottle | bus | car | cat | chair | cow | table | dog | horse | mbike | person | plant | sheep | sofa | train | tv | mean |
|---|---|---|---|---|---|---|---|---|---|---|---|---|---|---|---|---|---|---|---|---|---|
| conv4 flow | 28.2 | **34.1** | 20.4 | 17.1 | 50.6 | 36.7 | 20.9 | 19.6 | 15.7 | 25.4 | 12.7 | 18.7 | 25.9 | 23.1 | 21.4 | 40.2 | 21.1 | 14.5 | 18.3 | 33.3 | 24.9 |
| SIFT flow | 27.6 | 30.8 | 19.9 | 17.5 | 49.4 | 36.4 | 20.7 | 16.0 | 16.1 | 25.0 | 16.1 | 16.3 | 27.7 | **28.3** | 20.2 | 36.4 | 20.5 | 17.2 | 19.9 | 32.9 | 24.7 |
| NN transfer | 18.3 | 24.8 | 14.5 | 15.4 | 48.1 | 27.6 | 16.0 | 11.1 | 12.0 | 16.8 | 15.7 | 12.7 | 20.2 | 18.5 | 18.7 | 33.4 | 14.0 | 15.5 | 14.6 | 30.0 | 19.9 |
| Ours RN | 31.5 | 19.6 | 30.1 | 23.0 | 53.5 | 36.7 | 34.0 | 33.7 | 22.2 | 28.1 | 12.8 | 33.9 | 29.9 | 23.4 | 38.4 | 39.8 | 38.6 | 17.6 | 28.4 | 60.2 | 36.0 |
| Ours HN | 36.0 | 26.5 | 31.9 | 31.3 | 56.4 | **38.2** | 36.2 | 34.0 | 25.5 | 31.7 | **18.1** | 35.7 | 32.1 | 24.8 | 41.4 | 46.0 | 45.3 | 15.4 | 28.2 | 65.3 | 38.6 |
| Ours HN-ST | **37.7** | 30.1 | **42.0** | **31.7** | **62.6** | 35.4 | **38.0** | **41.7** | **27.5** | **34.0** | 17.3 | **41.9** | **38.0** | 24.4 | **47.1** | **52.5** | **47.5** | **18.5** | **40.2** | **70.5** | **44.0** |

Table 4: Per-class PCK on PASCAL-Berkeley correspondence dataset [4] ($\alpha = 0.1$, $L = \max(w, h)$).

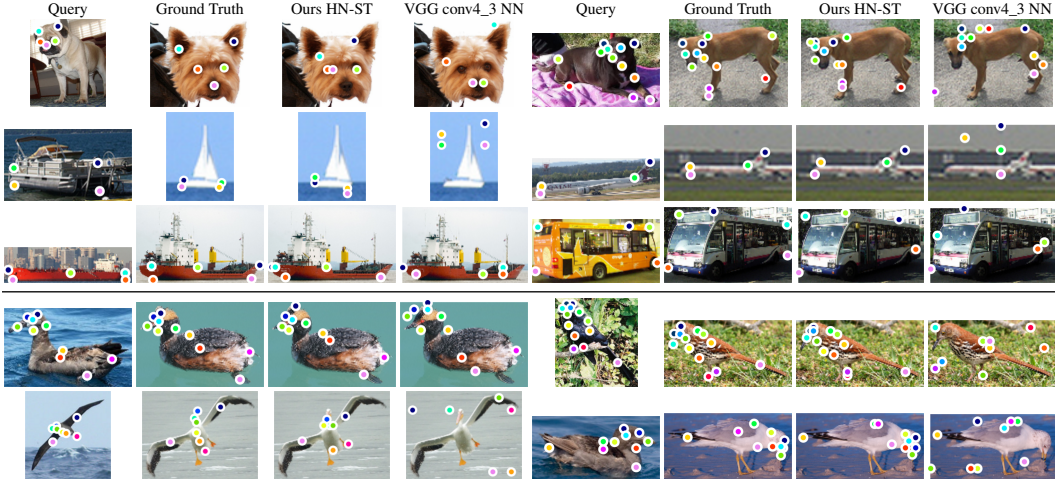

Figure 7: Qualitative semantic correspondence results on PASCAL [9] correspondences with Berkeley keypoint annotation [4] and Caltech-UCSD Bird dataset [30].

the feature space is more than 16 pixels away from the ground truth correspondence. We used the same architecture and training scheme for both datasets. Following convention [25], we measure PCK at 10 pixel threshold and compare with the state-of-the-art methods on Table 3. SIFT-flow [19], DaisyFF [31], DSP [18], and DM best [25] use additional global optimization to generate more accurate correspondences. On the other hand, just our raw correspondences outperform all the state-of-the-art methods. We note that the spatial transformer does not improve performance in this case, likely due to overfitting to a smaller training set. As we show in the next experiments, its benefits are more apparent with a larger-scale dataset and greater shape variations. Note that though we used stereo datasets to generate a large number of correspondences, the result is not directly comparable to stereo methods without a global optimization and epipolar geometry to filter out the noise and incorporate edges.

We also used KITTI raw sequences to generate a large number of correspondences, and we split different sequences into train and test sets. The details of the split is on the supplementary material. We plot PCK for different thresholds for various methods with densely extracted features on the larger KITTI raw dataset in Figure 5a. The accuracy of our features outperforms all traditional features including SIFT [22], DAISY [28] and KAZE [2]. Due to dense extraction at the original image scale without rotation, SIFT does not perform well. So, we also extract all features except ours sparsely on SIFT keypoints and plot PCK curves in Figure 5b. All the prior methods improve (SIFT dramatically so), but our UCN features still perform significantly better even with dense extraction. Also note the improved performance of the convolutional spatial transformer. PCK curves for geometric correspondences on individual semantic classes such as road or car are in supplementary material.

**Semantic Correspondence** The UCN can also learn semantic correspondences invariant to intra-class appearance or shape variations. We independently train on the PASCAL dataset [9] with various annotations [4, 36] and on the CUB dataset [30], with the same network architecture.

We again use PCK as the metric [32]. To account for variable image size, we consider a predicted keypoint to be correctly matched if it lies within Euclidean distance $\alpha \cdot L$ of the ground truth keypoint, where $L$ is the size of the image and $0 < \alpha < 1$ is a variable we control. For comparison, our definition of $L$ varies depending on the baseline. Since intraclass correspondence alignment is a difficult task, preceding works use either geometric [18] or learned [16] spatial priors. However, even our raw correspondences, without spatial priors, achieve stronger results than previous works.

As shown in Table 4 and 5, our approach outperforms that of Long et al.[21] by a large margin on the PASCAL dataset with Berkeley keypoint annotation, for most classes and also overall. Note that our

| mean | $\alpha = 0.1$ | $\alpha = 0.05$ | $\alpha = 0.025$ |
|---|---|---|---|
| conv4 flow[21] | 24.9 | 11.8 | 4.08 |
| SIFT flow | 24.7 | 10.9 | 3.55 |
| fc7 NN | 19.9 | 7.8 | 2.35 |
| ours-RN | 36.0 | 21.0 | 11.5 |
| ours-HN | 38.6 | 23.2 | 13.1 |
| ours-HN-ST | **44.0** | **25.9** | **14.4** |

Table 5: Mean PCK on PASCAL-Berkeley correspondence dataset [4] ($L = \max(w, h)$). Even without any global optimization, our nearest neighbor search outperforms all methods by a large margin.

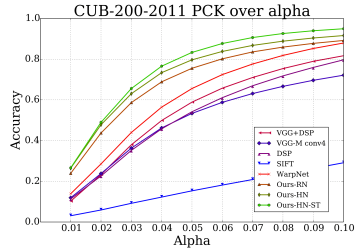

Figure 8: PCK on CUB dataset [30], compared with various other approaches including WarpNet [16] ($L = \sqrt{w^2 + h^2}$.)

| Features | SIFT [22] | DAISY [28] | SURF [3] | KAZE [2] | Agrawal et al. [1] | Ours-HN | Ours-HN-ST |
|---|---|---|---|---|---|---|---|
| Ang. Dev. (deg) | **0.307** | 0.309 | 0.344 | 0.312 | 0.394 | 0.317 | 0.325 |
| Trans. Dev.(deg) | 4.749 | 4.516 | 5.790 | 4.584 | 9.293 | **4.147** | 4.728 |

Table 6: Essential matrix decomposition performance using various features. The performance is measured as angular deviation from the ground truth rotation and the angle between predicted translation and the ground truth translation. All features generate very accurate estimation.

result is purely from nearest neighbor matching, while [21] uses global optimization too. We also train and test UCN on the CUB dataset [30], using the same cleaned test subset as WarpNet [16]. As shown in Figure 8, we outperform WarpNet by a large margin. However, please note that WarpNet is an unsupervised method. Please see Figure 7 for qualitative matches. Results on FlowWeb datasets are in supplementary material, with similar trends.

Finally, we observe that there is a significant performance improvement obtained through use of the convolutional spatial transformer, in both PASCAL and CUB datasets. This shows the utility of estimating an optimal patch normalization in the presence of large shape deformations.

**Camera Motion Estimation** We use KITTI raw sequences to get more training examples for this task. To augment the data, we randomly crop and mirror the images and to make effective use of our fully convolutional structure, we use large images to train thousands of correspondences at once.

We establish correspondences with nearest neighbor matching, use RANSAC to estimate the essential matrix and decompose it to obtain the camera motion. Among the four candidate rotations, we choose the one with the most inliers as the estimate $R_{pred}$, whose angular deviation with respect to the ground truth $R_{gt}$ is reported as $\theta = \arccos\left((\mathrm{Tr}\left(R_{pred}^{\top}R_{gt}\right) - 1)/2\right)$. Since translation may only be estimated up to scale, we report the angular deviation between unit vectors along the estimated and ground truth translation from GPS-IMU.

In Table 6, we list decomposition errors for various features. Note that sparse features such as SIFT are designed to perform well in this setting, but our dense UCN features are still quite competitive. Note that intermediate features such as [1] learn to optimize patch similarity, thus, our UCN significantly outperforms them since it is trained directly on the correspondence task.

# 5 Conclusion

We have proposed a novel deep metric learning approach to visual correspondence, that is shown to be advantageous over approaches that optimize a surrogate patch similarity objective. We propose several innovations, such as a correspondence contrastive loss in a fully convolutional architecture, on-the-fly active hard negative mining and a convolutional spatial transformer. These lend capabilities such as more efficient training, accurate gradient computations, faster testing and local patch normalization, which lead to improved speed or accuracy. We demonstrate in experiments that our features perform better than prior state-of-the-art on both geometric and semantic correspondence tasks, even without using any spatial priors or global optimization. In future work, we will explore applications for rigid and non-rigid motion or shape estimation as well as applying global optimization towards applications such as optical flow or dense stereo.

**Acknowledgments**

This work was part of C. Choy's internship at NEC Labs. We acknowledge the support of Korea Foundation of Advanced Studies, Toyota Award #122282, ONR N00014-13-1-0761, and MURI WF911NF-15-1-0479.

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
