[Supplementary Material · supplementary.pdf]

# Supplemental Materials for Universal Correspondence Network

## 1   Network Architecture

We use the ImageNet pretrained GoogLeNet [8], from the bottom conv1 to the inception_4a layer, but we used stride 2 for the bottom 2 layers and 1 for the rest of the network. We followed the convention of [6, 7] to normalize the features, which we found to stabilize the gradients during training. Since we are densely extracting features convolutionally, we implement q channel-wise normalization layer which makes all features have a unit L2 norm.

After the inception_4a layer, we place the correspondence contrastive loss layer which takes features from both images as well as the respective correspondence coordinates in each image. The correspondences are densely sampled from either flow or matched keypoints. Since the semantic keypoint correspondences are sparse, we augment them with random negative coordinates. When we use the active hard-negative sampling, we place the K-NN layer which returns the nearest neighbor of query image keypoints in the reference image.

We visualize the universal correspondence network on Fig. 1. The model includes the hard negative mining, the convolutional spatial transfomer, and the correspondence contrastive loss. The source code, pretrained network weights, caffe prototxt files and the interactive web visualization using [3] is available at `http://www.nec-labs.com/~mas/UCN/`.

## 2   Convolutional Spatial Transformer

The convolutional spatial transformer consists of a number of affine spatial transformers. The number of affine spatial transformers depends on the size of the image. For each spatial transformer, the origin of the coordinate is at the center of each kernel. We denote $x_i^s, y_i^s$ as the $x, y$ coordinates of the sampled points from the previous input $U$ and $x_i^t, y_i^t$ for $x, y$ coordinates of the points on the output layer $V$. Typically, $x_i^t, y_i^t$ are the coordinates of nodes on a grid. $\theta_{ij}$ are affine transformation parameters. The coordinates of the sampled points and the target points satisfy the following equation.

$$
\begin{pmatrix} x_i^s \\ y_i^s \end{pmatrix} = \begin{bmatrix} \theta_{11} & \theta_{12} \\ \theta_{21} & \theta_{22} \end{bmatrix} \begin{pmatrix} x_i^t \\ y_i^t \end{pmatrix}
$$

To get the output $V_i$ at $(x_i^t, y_i^t)$, we use bilinear interpolation to sample values $U$ around $(x_i^s, y_i^s)$. Let $U_{00}, U_{01}, U_{10}, U_{11}$ be the $U$ values at lower left, lower right, upper left, and upper right respectively.

$$V_i^c = \sum_n \sum_m U_{nm}^c \max(0, 1 - |x_i^s - m|) \max(0, 1 - |y_i^s - n|)$$
$$= (x_1 - x)(y_1 - y)U_{00} + (x_1 - x)(y - y_0)U_{10}$$
$$+ (x - x_0)(y_1 - y)U_{01} + (x - x_0)(y - y_0)U_{11}$$
$$= (x_1 - (\theta_{11}x_i^t + \theta_{12}y_i^t))(y_1 - (\theta_{21}x_i^t + \theta_{22}y_i^t))U_{00}$$
$$+ (x_1 - (\theta_{11}x_i^t + \theta_{12}y_i^t))(\theta_{21}x_i^t + \theta_{22}y_i^t - y_0)U_{10}$$
$$+ (\theta_{11}x_i^t + \theta_{12}y_i^t - x_0)(y_1 - (\theta_{21}x_i^t + \theta_{22}y_i^t))U_{01}$$
$$+ (\theta_{11}x_i^t + \theta_{12}y_i^t - x_0)(\theta_{21}x_i^t + \theta_{22}y_i^t - y_0)U_{11}$$

The gradients with respect to the input features are

$$\frac{\partial L}{\partial V_i^c}\frac{\partial V_i^c}{\partial U_{00}^c} = \frac{\partial L}{\partial V_i^c}(x_1 - x)(y_1 - y)$$
$$\frac{\partial L}{\partial V_i^c}\frac{\partial V_i^c}{\partial U_{10}^c} = \frac{\partial L}{\partial V_i^c}(x_1 - x)(y_0 - y)$$
$$\frac{\partial L}{\partial V_i^c}\frac{\partial V_i^c}{\partial U_{01}^c} = \frac{\partial L}{\partial V_i^c}(x_0 - x)(y_0 - y)$$
$$\frac{\partial L}{\partial V_i^c}\frac{\partial V_i^c}{\partial U_{11}^c} = \frac{\partial L}{\partial V_i^c}(x_0 - x)(y_1 - y)$$

Finally, the gradients with respect to the transformation parameters are

$$\frac{\partial V_i^c}{\partial \theta_{11}} = -x_i^t(y_1 - y)U_{00}^c - x_i^t(y - y_0)U_{10}^c$$
$$+ x_i^t(y_1 - y)U_{01}^c + x_i^t(y - y_0)U_{11}^c$$
$$\frac{\partial V_i^c}{\partial \theta_{12}} = -y_i^t(y_1 - y)U_{00}^c - y_i^t(y_1 - y)U_{10}^c$$
$$+ y_i^t(y - y_0)U_{01}^c + y_i^t(y - y_0)U_{11}^c$$
$$\frac{\partial V_i^c}{\partial \theta_{22}} = -x_i^t(x_1 - x)U_{00}^c + x_i^t(x_1 - x)U_{10}^c$$
$$- x_i^t(x - x_0)U_{01}^c + x_i^t(x - x_0)U_{11}^c$$
$$\frac{\partial V_i^c}{\partial \theta_{22}} = -y_i^t(x_1 - x)U_{00}^c + y_i^t(x_1 - x)U_{10}^c$$
$$- y_i^t(x - x_0)U_{01}^c + y_i^t(x - x_0)U_{11}^c$$

## 3   Additional tests for semantic correspondence

**PASCAL VOC comparison with FlowWeb**   We compared the performance of UCN with FlowWeb [10]. As shown in Tab. 1, our approach outperforms FlowWeb. Please note that FlowWeb is an optimization in unsupervised setting thus we split their data per class to train and test our network.

**Qualitative semantic match results**   Please refer to Fig 2 and 3 for additional qualitative semantic match results.

## 4   Additional KITTI Raw Results

We used a subset of KITTI raw video sequences for all our experiments. The dataset has 9268 frames which amounts to 15 minutes of driving. Each frame consists of Velodyne scan,

|  | aero | bike | boat | bottle | bus | car | chair | table | mbike | sofa | train | tv | mean |
|---|---|---|---|---|---|---|---|---|---|---|---|---|---|
| DSP | 17 | 30 | 5 | 19 | 33 | 34 | 9 | 3 | 17 | 12 | 12 | 18 | 17 |
| FlowWeb [10] | 29 | 41 | 5 | 34 | 54 | 50 | 14 | 4 | 21 | 16 | 15 | 33 | 26 |
| Ours-RN | 33.3 | 27.6 | 10.5 | 34.8 | 53.9 | 41.1 | 18.9 | 0 | 16.0 | 22.2 | 17.5 | 39.5 | 31.5 |
| Ours-HN | 35.3 | 44.6 | 11.2 | 39.7 | 61.0 | 45.0 | 16.5 | 4.2 | 18.2 | **32.4** | 24.0 | **48.3** | 36.7 |
| Ours-HN-ST | **38.6** | **50.0** | **12.6** | **40.0** | **67.7** | **57.2** | **26.7** | 4.2 | **28.1** | 27.8 | **27.8** | 45.1 | **43.0** |

Table 1: PCK on 12 rigid PASCAL VOC, as split in FlowWeb [10] ($\alpha = 0.05$, $L = \max(w, h)$).

stereo RGB images, GPS-IMU sensor input. In addition, we used proprietary segmentation data from NEC to evaluate the performance on different semantic classes.

| Scene type | City | Road | Residential |
|---|---|---|---|
| Training | 1, 2, 5, 9, 11, 13, 14, 27, 28, 29, 48, 51, 56, 57, 59, 84, | 15, 32, | 19, 20, 22, 23, 35, 36, 39, 46, 61, 64, 79, |
| Testing | 84, 91 | 52, 70, | 79, 86, 87, |

Table 2: KITTI Correspondence Dataset: we used a subset of all KITTI raw sequences to construct a dataset.

We excluded the sequence number 17, 18, 60 since the scenes in the videos are mostly static. Also, we exclude 93 since the GPS-IMU inputs are too noisy.

In Figure 5, we plot the variation in PCK at 30 pixels for various camera baselines in our test set. We label semantic classes on the KITTI raw sequences and evaluate the PCK performance on different semantic classes in Figure 4. The curves have same color codes as Figure 5 in the main paper.

# 5 KITTI Dense Correspondences

In this section, we present more qualitative results of nearest neighbor matches using our universal correspondence network on KITTI images on Fig. 6.

# 6 Sintel Dense Correspondences

In this section, we present more qualitative results of nearest neighbor matches using our universal correspondence network on Sintel images on Fig. 7.

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

Figure 1: Visualization of the universal correspondence network with the hard negative mining layer and the convolutional spatial transformer. The Siamese network shares the same weights for all layers. To implement the Siamese network in Caffe, we appended _p to all layer names on the second network. Each image goes through the universal correspondence network and the output features named $feature1$ and $feature2$ are fed into the K-NN layer to find the hard negatives on-the-fly. After the hard negative mining, the pairs are used to compute the correspondence contrastive loss.

Figure 2: Additional qualitative semantic correspondence results on PASCAL [4] correspondences with Berkeley keypoint annotation [1].

Figure 3: Additional qualitative semantic correspondence results on Caltech-UCSD Bird dataset [9].

Figure 4: PCK evaluations for semantic classes on KITTI raw dataset

Figure 5: PCK performance for various camera baselines on KITTI raw dataset.

Figure 6: Visualization of dense feature nearest neighbor matches on the KITTI dataset [5]. For each row, we visualize the query points (left) on the image $I_t$ at frame $t$ and the nearest neighbor matches (right) on the image $I_{t+1}$ at the next frame $t + 1$.

Figure 7: Visualization of dense feature nearest neighbor matches on the Sintel dataset [2]. For each row, we visualize the query points (left) on the image $I_t$ at frame $t$ and the nearest neighbor matches (right) on the image $I_{t+1}$ at the next frame $t+1$.