[Reviews · NeurIPS 2016]

Reviewer 1

Summary

This paper presents a deep learning approach to learn correspondences between pairs of images. They use a FCN with a novel correspondence contrastive loss and active hard-negative mining. They demonstrate the performance of this approach on a number of public datasets.

Qualitative Assessment

A well written description of the methods with convincing results. More details about the "semantic " aspect of this approach would be helpful.

Confidence in this Review

1-Less confident (might not have understood significant parts)


Reviewer 2

Summary

This paper presents a deep learning framework to obtain dense correspondences between a pair of natural images. The framework learns the correspondence map from the data and hence can be applied to both geometric (e.g. appearance change due to pose) as well as semantic correspondence (e.g. body regions across different birds). The key contribution compared to previous approaches include correspondence contrastive loss for efficient training, convolutional spatial transformer for local patch normalization as well as K-nearest neighbor layer for efficient correspondence search between feature vectors from 2 images. Paper is well written and experiments are performed on several relevant datasets with state-of-art results.

Qualitative Assessment

While there are a few details that can be investigate further, this paper does contain sufficient material to be considered for an oral presentation at NIPS. Comments: [1] Clarification on O(n) vs O(n^2) argument (forward passes)? The O(n^2) forward passes in the related papers are over image patches vs O(n) forward in this paper is over the image. Note the KNN layer in this paper also does O(mn) comparisons. Even in the related works, the features can be computed only once for every patch (forward pass) and the decision layers (last 2-3 layers) can be applied for all pairs. While the proposed framework would be computationally more efficient) in practice) with implicitly better book keeping and fully convolutional nature, I fail to understand how it has better computational complexity. [2] [Occlusions] Paper should include a discussion on how this framework handles occlusions. How the existing loss addresses that for certain point there should be no correspondence in the image (since the corresponding point in the second image is occluded)? During testing, the current metric (PCK curves) only penalizes for mis-registrations and hence occlusion is hope to addressed by choosing the right threshold. [3] References: Relevant reference (available on arxiv). Learning Dense Correspondence via 3D-guided Cycle Consistency. CVPR 2016.

Confidence in this Review

3-Expert (read the paper in detail, know the area, quite certain of my opinion)


Reviewer 3

Summary

The paper proposes an end to end network for correspondence matching. They introduce a modified version of the contrastive loss for their network and implement it in a fully convolutional manner which reduces the number of forward passes needed for training and testing the network. They also implement a hard negative mining algorithm for a faster training of the network. Authors also implement Convolutional Spatial Transformers which are layers that learn to do patch normalization for each activations in the ConvNet independently.

Qualitative Assessment

The paper has multiple contributions: -uses Fully Convolutional Networks for correspondence matching which reduces the computation complexity -introduces a modified version of contrastive loss for their application -introduces a hard negative mining approach which reduces training time -introduces a convolutional spatial transformer layer instead of patch normalization out of these 4 contribution I really like that they solve the problem using FCNs, rather than other approaches using patches The discussion on related work is fair and results are compared to various popular methods I consider the experiments sufficient for validating that the approach really works

Confidence in this Review

2-Confident (read it all; understood it all reasonably well)


Reviewer 4

Summary

Unlike previous patch similarity based methods, in this work the authors proposed a new model that discover visual correspondences using dense feature vector, correspondence contrastive loss function (a combination of mean squared loss and margin-based ranking loss), and CNN Spatial Transformer. The proposed method can successfully capture both geometric correspondence and semantic correspondence.

Qualitative Assessment

In this paper the authors proposed a universal correspondence network using CNN and pair-wise ranking loss. Overall this paper is well written and easy to follow. Below are the detailed review. 1. Equation(1) is the loss function used in the proposed model and s_i/(1-s_i) is acted as the identity function to select which part of the loss function will be used. In Line 121 the authors said they denote s = 0 for positive pairs and s=1 for negative pairs. In such way positive pairs will be trained by margin-based ranking loss and negative pairs will be trained by mean squared loss. This conflicts to Line 132 which claims that the mean squared loss is used for minimizing the distance between positive pairs and ranking loss is used to make sure negative pairs are at least m distance away. 2. In this work, all the key points are pre-selected so is it possible to run this algorithm without given any key points and let the model decide this? Basically this would become given a dense feature vector from left image, rank all vectors in the right image. Does training with hard negative pairs affect the performance in that case since a lot of vectors will not be updated during the training process? 3. The authors use a spatial transformer to mimic path normalization and show that by adding this the performance of the model increases. Does this differ from shortcuts? If so what is the advantage of spatial transformer over shortcut? 4. Figure 5 shows the Accuracy of PCK performance but no precision and recall. If I understand correctly, for a certain pixel threshold k, does the accuracy equal to #(actual correspondent pixels) / k? If so it would be interesting to see how the recall shape looks like.

Confidence in this Review

2-Confident (read it all; understood it all reasonably well)


Reviewer 5

Summary

This paper presents a novel architecture for learning the visual component correspondence between objects of the same category in different pictures. The fully convolution architecture makes use of a contrastive loss designed to efficiently reuse computation through the convolutional layers and a modified spatial transformer module to adaptively transform each input patch. Both features are novel and are crucial for the efficiency and effectiveness of the model.

Qualitative Assessment

The paper is well-written with clear explanations. The use of a fully convolutional architecture is well-justified as it greatly improves training efficiency compared to the triplet loss and the contrastive loss. The hard negative mining technique is essential in deep metric learning. There are some minor typos on lines 49 and in the caption for Figure 2. The authors used the term semantic correspondence numerous times. However, there is no evidence that the model learns any semantic meaning of parts of visual objects, but rather learns to extract abstract visual features that are shared across multiple instances of the same object part. The term geometric correspondence is more suitable. Experimental results are qualitatively unimpressive. In Figure 7, the HN-ST model fails to match the dog, boat and bus. This is further evidence that the model does not learn semantic correspondences since these three example pairs all have very different visual perspectives between the query and the test samples.

Confidence in this Review

2-Confident (read it all; understood it all reasonably well)